# Analyzing the Importance of Sensors for Mode of Transportation Classification [note 1]

**DOI:** 10.3390/s21010176

**Published:** 2020-12-29

**Authors:** Björn Friedrich, Carolin Lübbe, Andreas Hein

**Affiliations:** Department of Health Services Research, Medicine Faculty, Carl von Ossietzky University Oldenburg, Ammerländer Heerstraße 114-118, 26129 Oldenburg, Germany; carolin.luebbe@uni-oldenburg.de (C.L.); andreas.hein@uni-oldenburg.de (A.H.)

**Keywords:** mode of transportation classification, explainability, deep neural network, SHL challenge, feature visualization

## Abstract

The broad availability of smartphones and Inertial Measurement Units in particular brings them into the focus of recent research. Inertial Measurement Unit data is used for a variety of tasks. One important task is the classification of the mode of transportation. In the first step, we present a deep-learning-based algorithm that combines long-short-term-memory (LSTM) layer and convolutional layer to classify eight different modes of transportation on the Sussex–Huawei Locomotion-Transportation (SHL) dataset. The inputs of our model are the accelerometer, gyroscope, linear acceleration, magnetometer, gravity and pressure values as well as the orientation information. In the second step, we analyze the contribution of each sensor modality to the classification score and to the different modes of transportation. For this analysis, we subtract the baseline confusion matrix from a confusion matrix of a network trained with a left-out sensor modality (difference confusion matrix) and we visualize the low-level features from the LSTM layers. This approach provides useful insights into the properties of the deep-learning algorithm and indicates the presence of redundant sensor modalities.

## 1. Introduction

The broad acceptance of smartphones holds the potential for large-scale human-centered sensing and research. Smartphones contain a variety of different sensors for global localization and a body’s force. The data derived from smartphones enhances the research focused on the challenges arising with the growing number of people in urban and major metropolitan areas. One important challenge is traffic management in urban areas, since traffic congestion occurs naturally during rush hours. Information on people’s transport behavior can result in better routing and less congestion. Most smartphones can position themselves in a global frame of reference, e.g., GPS, but the accuracy depends on the signal quality and line of sight between the sensor and the satellites. The accuracy decreases significantly indoors or underground, as well as the features derived from the measurements. Inertial Measurement Units (IMU) are not reliant on external infrastructure. On the one hand, the data quality of the IMU does not depend on whether the sensor is underground or not, and on the other hand, the IMU data depend on the kinematic chain between the sensor and the source of the force applied to the sensor.

As part of the SHL recognition challenge 2020 [1], we proposed a deep-learning-based algorithm that combines augmentation and LSTM layers as well as several convolutional, and fully connected layers to perform transportation mode classification using IMU data from smartphone sensors [2].

However, decisions made by deep neural networks are difficult to understand and interpret due to their black box character. Explanatory Artificial Intelligence (XAI) tackles this problem and allows more transparent decisions that can be explained in a certain level of detail. This is essential since explanations can be used to ensure algorithmic fairness, identify potential bias, and problems in the training data, and to validate that the algorithms work as expected [3]. Compared to deep neural networks for image classification, where learned features can be visualized more intuitively, and thus be interpreted more easily by humans, visualizations in the time series domain are challenging. This is since the input as well as the features are more abstract and include a time dimension. In this context, this paper targets the basic understanding of what a deep neural network learns and which inputs have the greatest influence on accuracy. For this purpose, we trained our network for the SHL Challenge 2020 with a leave-one-sensor-out strategy and computed the difference confusion matrices of the network and the baseline trained with all sensors. Moreover, we used autoencoders to visualize the low-level features learned by the LSTM layers of each sensor. By combining the two results, we were able to identify the contribution of individual sensors to classification accuracy and detect redundancies.

The paper is structured as follows: Section 2 gives a brief overview of the state of the art. Afterwards, in Section 3 we provide details regarding the used dataset, the pre-processing pipeline and the used algorithm. Section 4 summarizes the results of our analysis and we discuss the obtained results in the final Section 5.

## 2. State of the Art

For several years extensive work on understanding and sensing the mobility behavior of people has been carried out. This section first introduces the state of the art regarding mode of transportation classification using machine learning approaches based on smartphone sensor data. Subsequently, we examine explanatory visualization techniques that provide a better understanding of deep neural networks and their decisions, as well as methods that can be used to shed light on the influence of inputs.

All approaches that include contextual information are not considered, since this research focuses on the use of information derived from smartphone sensors. A common approach is to understand the detection of the mode of transportation as a classification problem. We have assigned related works to the following two categories: 1. traditional machine learning-based classification and 2. deep-learning-based classification. Antar et al. [4] and Liono et al. [5] proposed random forest (RF) classifiers that achieved an accuracy of 92% and 91% on the SHL dataset and Crowdsignals dataset. Yu et al. [6] extracted features from three sensors (accelerometer, magnetometer and gyroscope) and proposed support vector machines (SVM) as the best classifier for detecting a person’s mode of transportation (i.e., standing still, walking, running, cycling, and in the vehicle). Similar findings were also made by Fang et al. [7]. Another traditional approach only based on acceleration data was proposed by Hemminski et al. [8] to detect five different modes of transportation (i.e., bus, train, underground, tram and car). Recently, large-scale datasets have been made available which enable the application of deep-learning techniques. The deep-learning algorithms are outperforming the traditional approaches which are using handcrafted features. Jeyakumar et al. [9] proposed a deep convolutional bidirectional-LSTM ensemble trained directly on raw sensor data on the SHL dataset. Using this approach, an F${}_{1}$-score on 96% was achieved for transportation mode classification. Qin et al. [10] introduced a deep-learning-based algorithm that combines a CNN and LSTM network. By using CNN-extracted and handcrafted features (i.e., segment and peak features), the algorithm can distinguish the transportation modes with an accuracy of 98.1% on the SHL dataset. Vu et al. [11] proposed a gate-based recurrent neural network to detect the transportation mode on the HTC dataset. This accelerometer-based approach achieved an accuracy of 94.72%. Tambi et al. [12] presented a CNN that distinguishes four transportation modes (bus, car, subway, train) by using mobile sensor data derived from an accelerometer and a gyroscope in the spectral domain. An accuracy of 91% was achieved.

Although there is a lot of work done on the development and modification of LSTM architectures, the decisions made by deep neural networks are still difficult to understand and interpret, due to their black box character. To provide a better understanding, different explanatory techniques have been proposed.

One technique that visualizes decision-making in CNNs is the Class Activation Map (CAM) [13]. It indicates the discriminatory image region used to identify a specific class. Grad-CAM is a more versatile version of CAM. Using gradients applied to the last convolution layer of a CNN, Grad-CAM tries to find salient regions in the input space [14]. However, CAM is not only applied for a deeper understanding of the decision-making process for image classification, but also for the classification of time series. In this context, Wang et al. [15] introduced a one-dimensional CAM that highlights class-specific regions that have contributed most to a particular classification of time series. This method gives insights into the properties of the deep-learning algorithm or its decision-making process, but does not provide the possibility to identify redundant features.

Several works have focused on interpreting the hidden states of LSTMs or hidden layers of CNNs. Karpathy et al. [16] showed the existence of interpretable cells in LSTMs that kept track of dependencies, such as line length, quotes and brackets in character-level language models. Moreover, the hidden state of LSTMs on different inputs can be explored interactively by the visual analysis tool LSTMVis [17] for recurrent neural networks. To intercept the hidden layers of deep neural networks, Moreira et al. recently employed autoencoders to provide information for the interpretation of classifiers, and to enable the investigation of misclassifications in the dataset from emerging clusters [18]. These methods offer the possibility to increase the explainability of the functionality of models, whereas in this paper we mainly focused on the explainability of the input and on the low-level features. Another area of research is feature selection, which not only aims to gain a better understanding of the features, but also to improve the prediction accuracy and speed of classification. With the intention of improving prediction accuracy, Liu et al. [19] proposed a *leave-one-feature-out wrapper* method. The *leave-one-covariate-out* method [20] aims at estimating the importance of local features. Furthermore, Azarbayejani et al. [21] introduced an approach for the evaluation of the redundancy of sensor networks, which is based on a *leave-one-sensor-out* analysis. These methods, which can be applied in a straightforward manner, improve the explainability of the features used and allow identification of redundant features or sensor modalities. At this point we see the potential to extend these methods and to introduce them into the area of explainability.

## 3. Materials and Methods

The provided part of the Sussex–Huawei Locomotion-Transportation (SHL) dataset [22,23] contains data from smartphones carried on the body in various positions. The dataset was collected with three participants over 31.6 d, each of them carrying four phones positioned at the four different locations *hand, bag, hips*, and *torso*. The values of the hardware sensors *accelerometer, gyroscope, magnetometer*, and *pressure*, as well as the software sensor values of *linear acceleration*, *gravity* and *orientation*. A virtual, i.e., software sensor is constructed by using the values of one or more hardware sensors to compute the value of the software sensor. The measurement frequency was 100 Hz. Each individual sensor value was labelled, i.e., 100 labels are available for 1 s.

The dataset includes eight different modes of transportation *still, walk, run, bike, car, bus, train,* and *subway*. The samples are consecutive in time for the training and validation set, as opposed to the test set. The training data comprises the values of all four phone locations from one participant. The validation data comprises of the values of the other two participants from all locations as well. The test set contains data from the same users in the validation set, but only from one unknown phone location. Overall, there are 196,072 training samples, 28,789 validation samples, and 57,573 test samples. Moreover, the dataset has a large class imbalance. The Figure 1 and Figure 2 show the label distribution in the training, validation and challenge test set.

### 3.1. Pre-Processing

Before pre-processing, we performed some data integrity checks. We found that the labels for some samples are not uniform, i.e., the samples contain transitions of modes of transportation. Since the number of samples containing a transition was less than 1%, we assigned the label by majority decision. Thus, our dataset has only one label instead of 500 for each sample. Then, the training set has been merged with the validation set. To overcome the class imbalance, we followed a simple approach and oversampled by copying random samples and undersampled by deleting random samples. We used 30,000 samples, because the number of classes in which samples had to be deleted equals the number of classes in which samples had to be copied. Before balancing, the full dataset was split into new training, validation, and test sets in a stratified way. Since the samples in the challenge test set are not in a consecutive order, the samples were chosen at random. 75% of the full dataset was assigned to the training set, 15% to the validation set, and the remaining 10% to our private test set. Finally, the data from all phone locations were merged. The training set contains 720,000 samples, the validation set 144,000 samples, and the test set 96,000 samples.

Two pre-processing steps were applied on the balanced dataset. The first one was to apply a low-pass filter on all data. We used a second order filter with a cut-off frequency of 25 Hz. The second step was standard scaling by subtracting the mean and dividing by the variance. Standard scaling was applied to each feature in each dimension separately. Augmentation was applied with a probability of 50% during runtime. Figure 3 shows the difference in the acceleration between a raw sample and the augmented sample of the class *still*.

2

The activations of the LSTM layers were not preprocessed. The activation function of the LSTM layers were *tanh*, and therefore the output was already scaled to [−1, 1]. The output dimension were (batchsize,500,64), and the last 64 values of the output sequences were used. For visualization we transformed each encoded value by
(1)$$y=sign\left(x\right)\times \ln\left(|x|+1\right)$$
where sign denotes the signum function, ln the natural logarithm and *x* the input variable. The natural logarithm is not defined for 0, but if the activation is −1 the input variable to the natural logarithm is 0. Our implementation returns the input value if the input value is so small that the natural logarithm function cannot compute the result. This happens for 0 and for very small values close to 0, because computers have only a limited number of bytes for storing values (floating point arithmetic).

### 3.2. Algorithm

For finding the best architecture, we started off with a very small neural network and followed a Greedy approach. We subsequently added layers and adjusted parameters. If the result improved, the adjustments were kept, if the result was worse, the adjustments were reverted. The architecture, we propose (Figure 4) combines an augmentation and an LSTM layer, as well as several convolutional and fully connected layers to perform transportation mode classification. The input data is split into seven streams, one stream per sensor. To artificially increase the number of training samples, an augmentation layer is implemented, which augments four windows of size 50 of each sample with a factor of 2. This is followed by an LSTM layer that can store information about time to find temporal correlations of the input sequences. The LSTM layer comprises 64 neurons, sigmoid recurrent activation and tanh activation. It is followed by a dropout layer, with a dropout rate of 0.25, that is used to avoid overfitting, a convolution layer, and at the end of each stream a maximum pooling layer. The convolutional layer consists of 128 filters, a kernel size of 8, stride length of 2 and a Leaky ReLU activation function with α=0.001. Maximum pooling was performed with stride length of 2. Then, the seven streams are merged via a concatenation layer, which allows us to combine all features to extract meaningful information. Afterwards, a convolutional layer and a maximum pooling layer are used 4 times in a row, whereupon a flatten layer completes the second block (see Figure 5). In all type 2 blocks, maximum pooling, the convolutional stride and the Leaky ReLU activation with α=0.001 were the same. The number of filters and the filter size were arranged in ascending order 64, 64, 128, 128, and 16, 32, 64, 64. The subsequent fully connected layers, each followed by a dropout layer, recombine the representations learned from the convolution layer and reduce the dimension. Both blocks of type 3 used the same parameters. The dense layer had 256 neurons, the dropout rate was 0.25 and Leaky ReLU was used as activation function, as before. In the last step, the classification layer uses the SoftMax activation function for the mode of transport classification. We used categorical cross-entropy loss and the F${}_{1}$ score as metric. The Adam optimization algorithm was used for gradient optimization and we used a learning rate schedule with exponential decay after the first 10 epochs with an initial learning rate of 0.001.

For dimensionality reduction and visualization we used a common autoencoder. The basic idea of an autoencoder is to find the best representation of high-dimensional data in a low-dimensional latent space. The best latent space representation is the representation, where the input can be reconstructed with a minimal error. The autoencoder was trained on all last activations of the LSTM layer of all samples. For each sensor modality a separate autoencoder was trained. The autoencoders were comprised of five layers with 600, 150, 2, 150, 600 neurons. The architecture is shown in Figure 6. The autoencoder is reducing the dimensionality from 600 dimensions to two. The upper part of the network is the encoder and the lower part the decoder. The latent layer with two neurons and the output layer were activated by a linear activation function and all other layers by the ReLU function. The used optimizer was Adam with a learning rate of 0.001, and the mean squared error as loss function. The number of training epochs was not uniform, because we used the early stopping criteria with a minimum delta of 0 for the validation loss, and a patience of 5 epochs, i.e., the training was stopped after 5 epochs without any improvements.

### 3.3. Difference Confusion Matrix

The difference confusion matrices are computed by subtracting the confusion matrix of the network trained with all sensor modalities from the confusion matrix of the network trained without a certain sensor modality. A positive value in a cell means that the value is larger for the network without one sensor modality. A negative value in a cell means that the value is smaller for the network without one sensor modality and 0 means the values are equal. A positive value on the diagonal means that the network without one sensor modality is better classifying the corresponding class and a negative value means that the network without one sensor modality is worse in classifying the corresponding class. A positive value in a non-diagonal cell means the network without one sensor modality is worse in distinguishing the corresponding classes and a negative value means it is better in distinguishing. A value of 0 means equal classification performance.

## 4. Results

After some preliminary experiments, we found that the model has difficulties distinguishing between the classes *train* and *subway*. Therefore, we put a higher weight (3x) on the gradient update for the class *train*. The Figure 7 and Figure 8 show the graphs of the F${}_{1}$ score and the loss of the final training. In the beginning the score and the loss have a high slope and later the slope is asymptotically approaching the limit 0. During the first 10 epochs the validation score is slightly better than the training score and the validation loss is slightly smaller than the training loss. The confusion matrix shows that the model performs best on the classes *walk* and *run* and worst on the classes *still* and *subway*. An overview of the best epochs, the score on our private test set, and on the challenge test set is given in Table 1 and Table 2. The best epoch was epoch 77 with a validation score of 98.93% and a score of 98.96% on our private test set. The largest score difference for our private test set (0.77%) is obtained by subtracting the score of the network trained without pressure from the score of the network trained with all sensor modalities. Furthermore, the largest score difference for the challenge test set (8.33%) is found by subtracting the score of the network trained without orientation from the score of the network trained without linear acceleration. Only two networks with one left-out sensor, gyroscope and orientation, have a worse score on the challenge test set than the network trained with all sensor modalities.

The difference confusion matrices are shown in Table 3, Table 4, Table 5, Table 6, Table 7, Table 8, Table 9 and Table 10 and the plots of the encoded last activations from the LSTM layers are shown in Figure 9, Figure 10, Figure 11, Figure 12, Figure 13, Figure 14 and Figure 15. The color codes are blue for the class *still*, orange for the class *walk*, green for the class *run*, red for the class *bike*, violet for the class *car*, brown for the class *bus*, pink for the class *train*, and grey for the class *subway*.

The diagonal values of the difference confusion matrix without acceleration are negative, except for class *bus* (min. −98, max. 12). The maximum difference is found in the cell *(train, train)* and the minimum difference in the cell *(run, run)*. Eleven of the remaining 56 non-diagonal cells contain a negative value (min. −18, max. −1), nine cells contain 0 and 36 cells contain positive values (min. 1, max. 55). The plot of the last activations of the acceleration LSTM layer shows that the activations of the classes *bike* (red) and *run* (green) overlap the least. The activations of the class *walk* (orange) are not overlapping in the area centered at (0, 1). Acceleration is an important feature for classifying *run* and *bike*, but that the largest drop in accuracy is found for class *train*. In contrast, more bus samples (12) were classified correctly without the accuracy features. Considering the scores in Table 1, we see that the loss in the score on our private test set is the second smallest loss (0.19%) and there is a very small increase of 0.03% in the score on the challenge test set.

The diagonal values of the difference confusion matrix without gravity are negative except for the class *bike* (min. −86, max. 1). The maximum difference is found in the cell *(train, train)* and the minimum difference in the cell *(bike, bike)*. Fourteen of the remaining 56 non-diagonal cells contain a negative value (min. −21, max. −1), eight cells contain 0 and 34 cells contain positive values (min. 1, max. 47). The plot of the activations shows overlapping activations for the gravity LSTM layer for all classes. The plot of the activations of the gravity LSTM layer shows that the least overlapping activations are of classes *bike* (red) and *run* (green). The activations of the class *walk* (orange) are non-overlapping in the area centered at (0, 1). The features of the layer are overlapping for all classes. Accordingly, the difference confusion matrix (Table 5) is the only one that has a positive value on the diagonal. The score on the challenge test set increases by 3.69% and the score on the private test set decreases by 0.31%.

The diagonal values of the difference confusion matrix without gyroscope are all negative (min. −104, max. −11). The maximum difference is found in the cell *(train, train)* and the minimum difference in the cell *(run, run)*. Ten of the remaining 56 non-diagonal cells contain negative values (min. −15, max. −1), seven cells contain 0 and 39 cells contain positive values (min. 1, max. 66). The plot of the activations of the gyroscope LSTM layer shows that the least overlapping activations are of classes *bike* (red) and *run* (green). The activations of the class *walk* (orange) are non-overlapping in the areas centered at (0, −0.5), (−1, −0.5) and (0, −1). Thus, the gyroscope is one sensor that is redundant for the classes *run* and *bike*. The difference confusion matrix in Table 6 and the largest drop in performance is found in the classes *train* and *subway* and not in *run* and *bike*. However, the gyroscope is important for the other classes, because the test score has the second largest decrease of 0.74% for our private test set. The same holds for the challenge test set (0.29%).

The diagonal values of the difference confusion matrix without linear acceleration are negative, except for the class *bike* (0). The maximum difference is found in the cell *(train, train)* and the minimum difference in the cell *(run, run)*. Sixteen of the remaining 56 non-diagonal cells contain negative values, eight cells contain 0 and 42 cells contain positive values (min. 1, max. 59). The effect of leaving out the linear acceleration is similar to leaving out the acceleration. The plots in Figure 9 and Figure 12 look similar and the distribution of the loss in accuracy in Table 4 and Table 7 is similar as well. However, the score on the challenge test set is the highest score for all left-out sensors and increases by 7.32% and the score on the private test set decreases by 0.28%.

The diagonal values of the difference confusion matrix without magnetometer are all negative. The maximum difference is found in the cell *(still, still)* and the minimum difference in the cell *(run, run)*. Five of the remaining 56 non-diagonal cells contain negative values (min. −10, max. −1), eleven cells contain 0 and 40 cells contain positive values (min. 1, max. 92). The plot of the activations for the magnetometer LSTM layer shows the least overlapping activations are of classes *subway* (grey) and *train* (pink). A cluster of activations of class *run* (green) can be found centered around (−0.4, 1.25) and clusters of activations of class *bike* (red) can be found at (−1, 0.75) and (0.75, 0.75). The plot indicates that the magnetometer is important for the classes *train* and *subway*. Considering the difference confusion matrix Table 8 reveals that the second and third largest loss in accuracy can be found in those two classes. The largest loss in accuracy is in class *still*, even though only a few features of class *still* are visible in the plot ((−0.25, −1.0), (−0.4, 0.6), (−0.4, 1.2), (0.0, 1.2), and (0.4, 0.8)).

The diagonal values of the difference confusion matrix without orientation are all negative. The maximum difference is found in the cell *(train, train)* and the minimum difference in the cell *(run, run)*. Ten of the remaining 56 non-diagonal cells contain negative values (min. −15, max. −1), seven cells contain 0 and 39 cells contain positive values (min. 1, max. 108). The plot of the activations shows overlapping activations for the orientation LSTM layer for all classes and does not show any substantial contribution of the orientation to any class, but the difference confusion matrix shows a large loss in accuracy for the classes *train* and *subway*. Comparing the losses in performance in Table 1 shows that the overall losses are moderate with 0.51% and 1.01% for the private test set and the challenge test set, respectively.

The diagonal values of the difference confusion matrix without pressure are all negative. The maximum difference is found in the cell *(subway, subway)* and the minimum difference in the cell *(run, run)*. Five of the remaining 56 non-diagonal cells contain negative values (min. −6, max. −1), nine cells contain 0 and 44 cells contain positive values (min. 1, max. 167). The plot of the activations of the pressure sensor (Figure 15) is remarkable because the structure is totally different to the other sensor activation plots. All other activations are starting around (0, 0) and then evolve in all directions, whereas the pressure activations look like a line with bulges. The lower right part of the plot shows that the pressure features are useful to distinguish *bike, car*, and *train*. The difference confusion matrix Table 10 is contradicting the plot. The false classifications of these three classes differ slightly compared to using the sensor. The largest loss in performance is found in the classes *train* and *subway*. Leaving out the pressure sensor results in a loss in accuracy by 0.85% on the private test set and an increase in accuracy by 4.88% on the challenge test set.

## 5. Discussion and Conclusions

Considering all results, the classes *train* and *subway* are most affected by removing one sensor modality. In six out of seven cases, these two classes have the highest loss in performance. The sensors acceleration, gyroscope, and linear acceleration are redundant for the two classes *run* and *bike* and the least important sensor seems to be the gravity sensor. Furthermore, the pressure sensor seems to be the most important sensor, according to Table 2 and the shape of the activations in Figure 15. The results also showed that the software sensors linear acceleration and orientation do not give substantial contribution to the performance. The network can internally learn the important information from the hardware sensors. Moreover, the difference confusion matrices and the activation plots helped to identify redundancies regarding the sensors and certain classes. Even though the plots are supporting the findings in the confusion matrices, the use of the plots is limited. The plots are useful to visualize the magnitude of activation of the different classes and the general structure of the plots can be used to identify sensors that should be investigated further. However, we showed that the difference confusion matrices are applicable in cases where visualization methods are only partially useful.

Our contribution to explainable machine learning is the introduced difference confusion matrices as a tool for analyzing deep neural networks. We showed that the insights match the visualization and that the difference confusion matrices can be used when visualization is limited. We also identified sensor redundancies and revealed that the network internally learns most of the information provided by the software sensors.

## Figures and Tables

**Figure 1 sensors-21-00176-f001:**
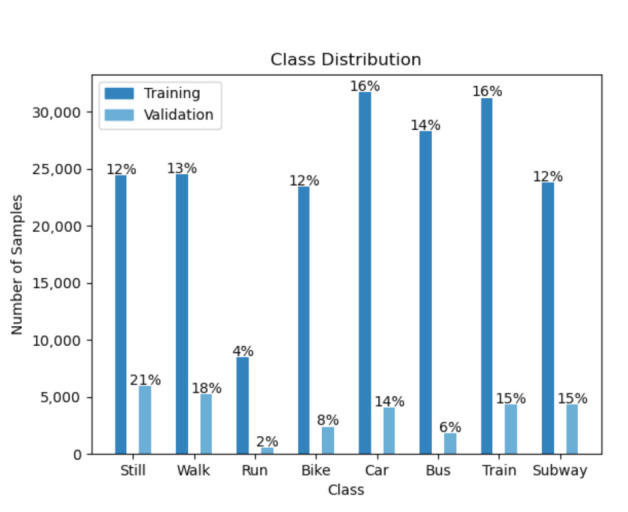
The histogram of the distribution of the labels in the training and validation set.

**Figure 2 sensors-21-00176-f002:**
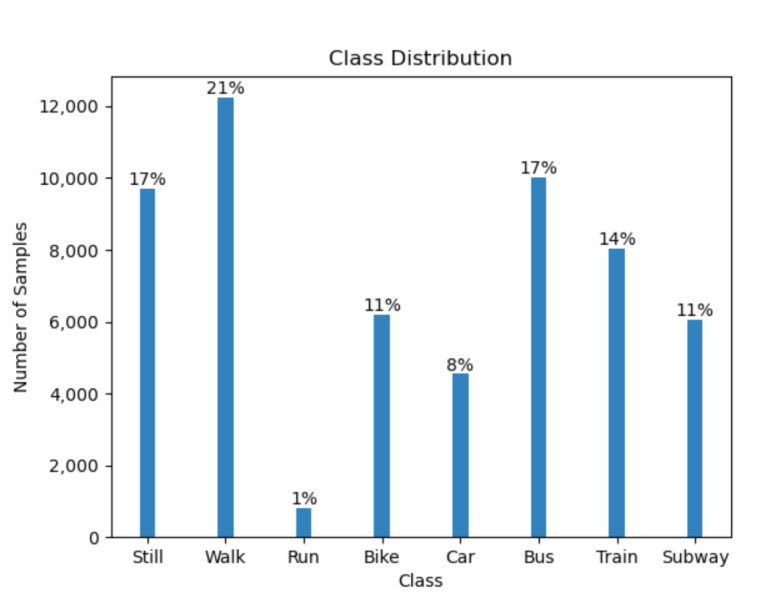
The histogram of the distribution of the labels in the challenge test set.

**Figure 3 sensors-21-00176-f003:**
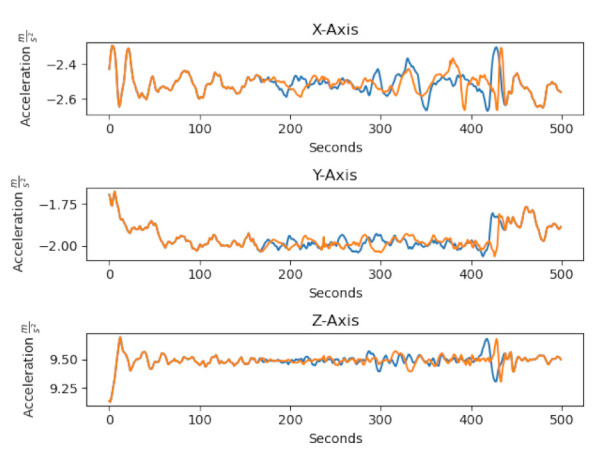
The difference between the raw sample and the augmented sample. The blue line is the raw sample and the orange line the augmented sample.

**Figure 4 sensors-21-00176-f004:**
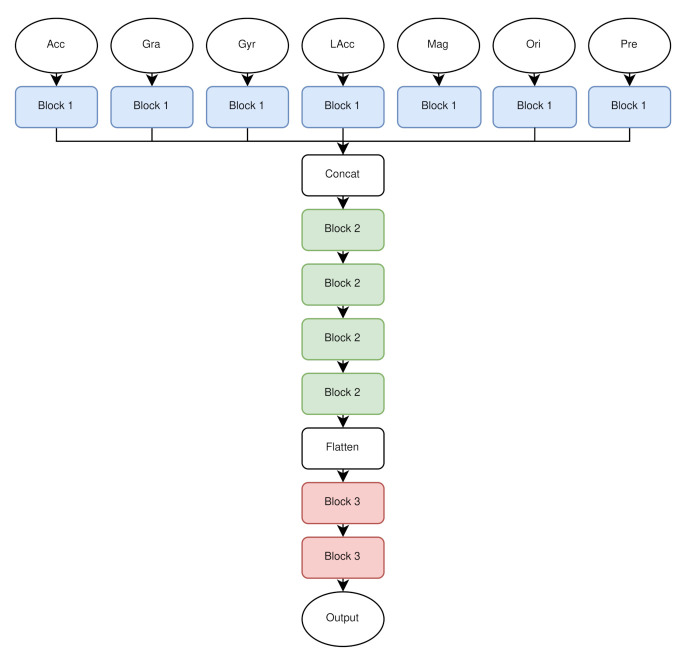
The architecture of the model. Each sensor modality had its own input, and the intermediate features were fused in the concatenation layer in the second dimension.

**Figure 5 sensors-21-00176-f005:**
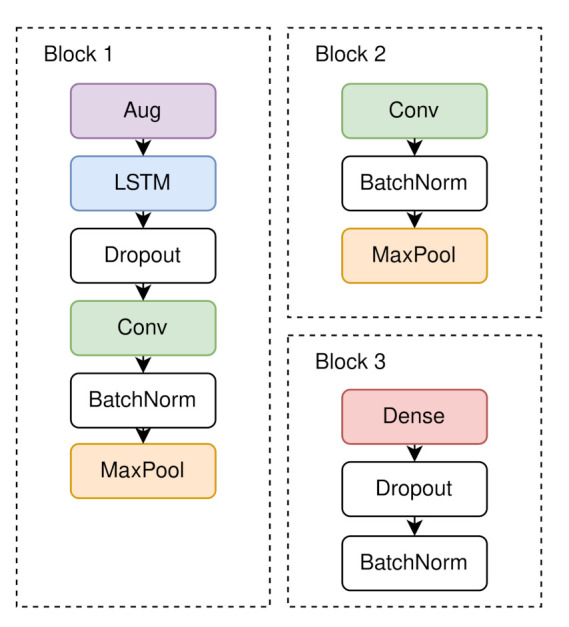
A detailed view of the three different blocks of layers used in our architecture.

**Figure 6 sensors-21-00176-f006:**
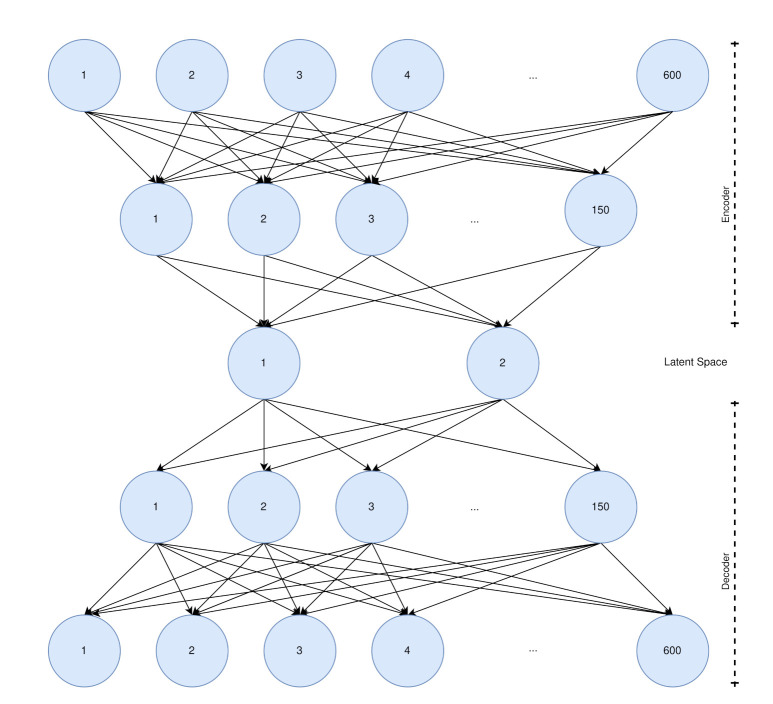
The autoencoder architecture used for dimensionality reduction. The first neuron of the latent space is the *x*-value and the second neuron of the latent space is the *y*-value.

**Figure 7 sensors-21-00176-f007:**
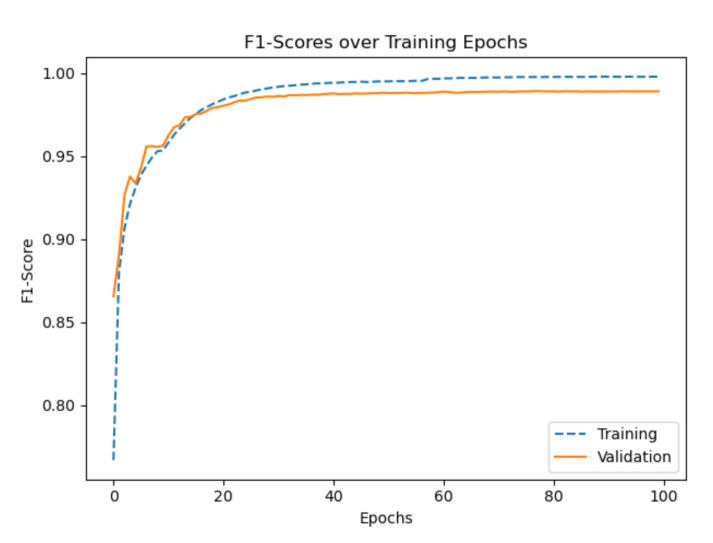
The progress of the score for the final training for 100 epochs. The progress shows an asymptotic behavior after around about 40 epochs.

**Figure 8 sensors-21-00176-f008:**
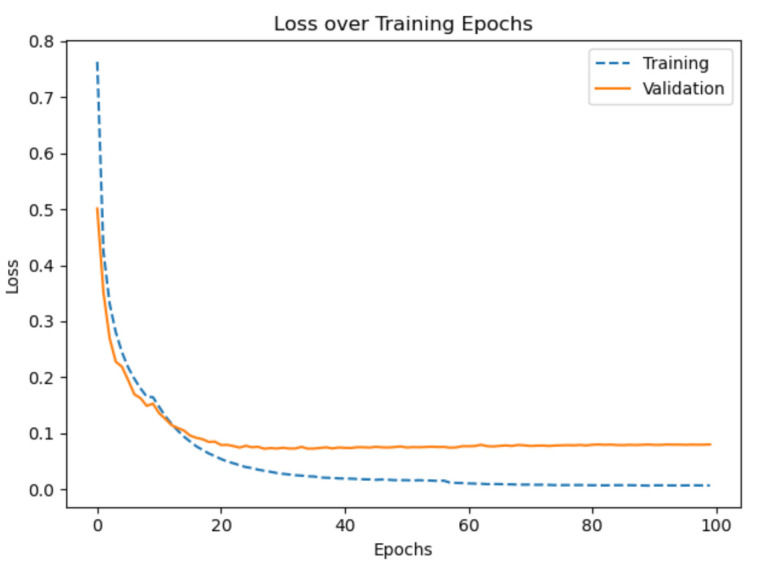
The progress of the loss for the final training for 100 epochs. The progress shows an asymptotic behavior after around about 40 epochs. The progress corresponds to the progress of the score.

**Figure 9 sensors-21-00176-f009:**
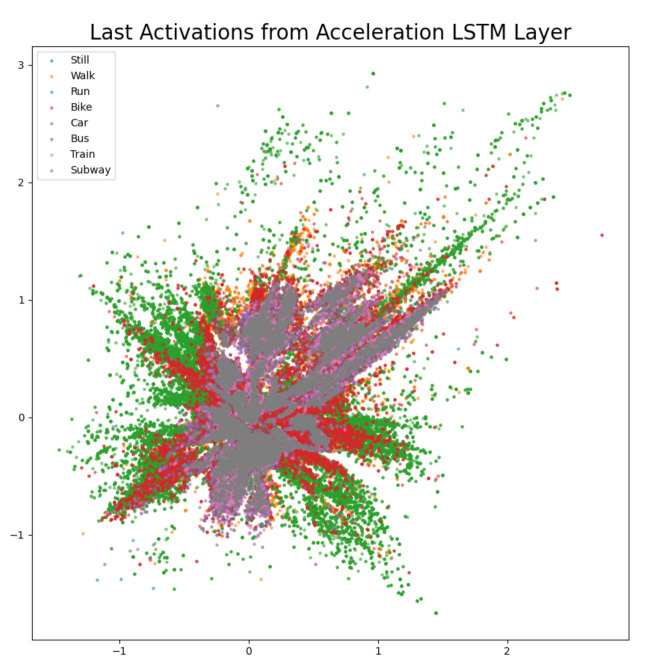
The 2D encoded representation of the last activations of the LSTM layer for the acceleration sensor. Color codes: blue (still), orange (walk), green (run), red (bike), violet (car), brown, (bus), pink (train), grey (subway).

**Figure 10 sensors-21-00176-f010:**
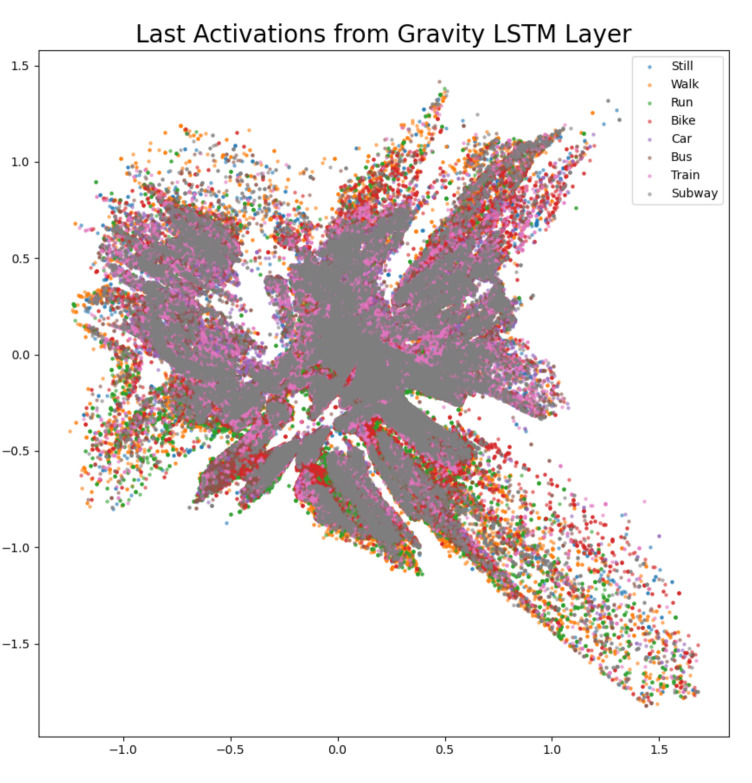
The 2D encoded representation of the last activations of the LSTM layer for the gravity software sensor. Color codes: blue (still), orange (walk), green (run), red (bike), violet (car), brown, (bus), pink (train), grey (subway).

**Figure 11 sensors-21-00176-f011:**
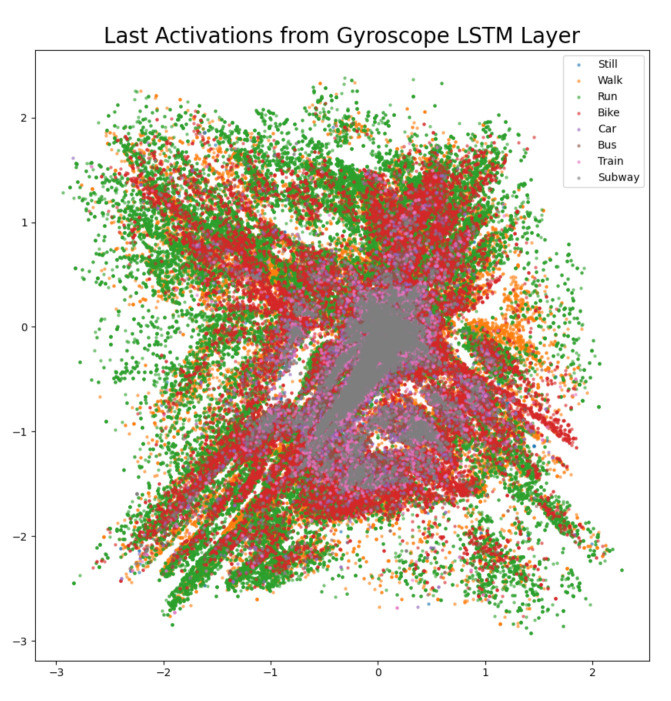
The 2D encoded representation of the last activations of the LSTM layer for the gyroscope. Color codes: blue (still), orange (walk), green (run), red (bike), violet (car), brown, (bus), pink (train), grey (subway).

**Figure 12 sensors-21-00176-f012:**
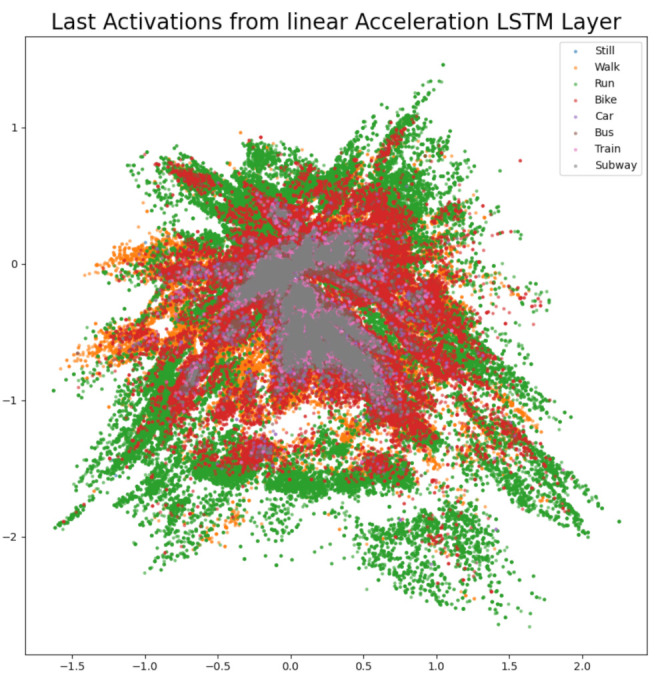
The 2D encoded representation of the last activations of the LSTM layer for the linear acceleration software sensor. Color codes: blue (still), orange (walk), green (run), red (bike), violet (car), brown, (bus), pink (train), grey (subway).

**Figure 13 sensors-21-00176-f013:**
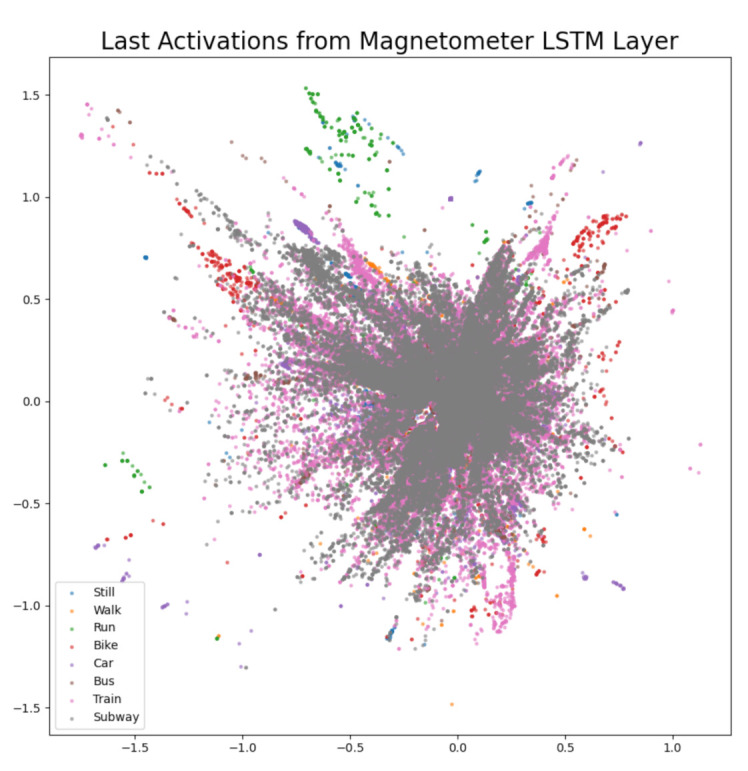
The 2D encoded representation of the last activations of the LSTM layer for the magnetometer. Color codes: blue (still), orange (walk), green (run), red (bike), violet (car), brown, (bus), pink (train), grey (subway).

**Figure 14 sensors-21-00176-f014:**
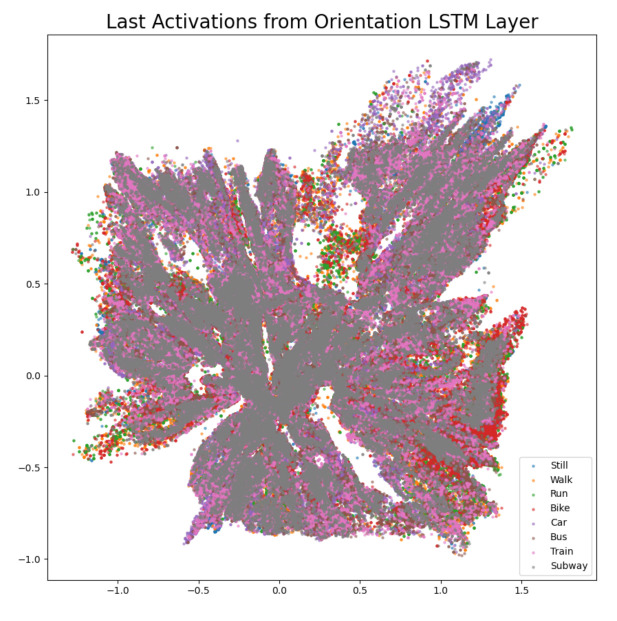
The 2D encoded representation of the last activations of the LSTM layer for the orientation software sensor. Color codes: blue (still), orange (walk), green (run), red (bike), violet (car), brown, (bus), pink (train), grey (subway).

**Figure 15 sensors-21-00176-f015:**
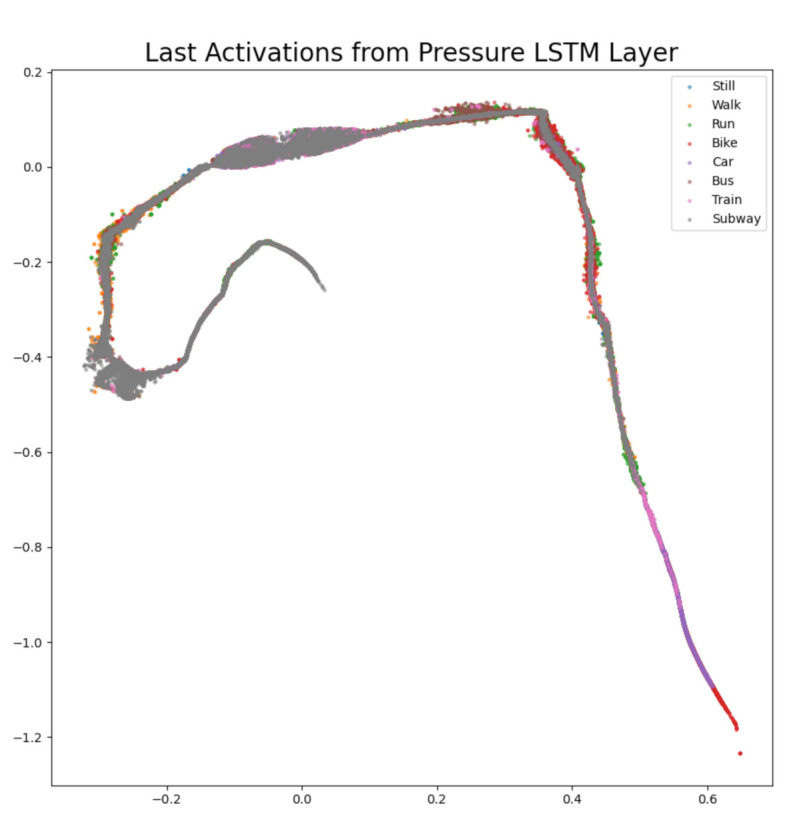
The 2D encoded representation of the last activations of the LSTM layer for the pressure sensor. Color codes: blue (still), orange (walk), green (run), red (bike), violet (car), brown, (bus), pink (train), grey (subway).

**Table 1 sensors-21-00176-t001:** An overview about the scores and best epochs of the networks trained with one sensor modality left-out. For the sake of comparison, the first row shows the baseline values of the network trained with all sensor modalities.

Left-Out Sensor	Best Epoch	Validation Score	Private Test Score	Challenge Score
None	77	98.93%	98.96%	52.80%
Acceleration	50	98.73%	98.77%	52.83%
Gravity	50	98.58%	98.65%	56.49%
Gyroscope	50	98.21%	98.22%	52.51%
linear Acceleration	49	98.66%	98.68%	60.12%
Magnetometer	50	98.76%	98.80%	55.95%
Orientation	50	98.42%	98.45%	51.79%
Pressure	50	98.16%	98.11%	57.68%

**Table 2 sensors-21-00176-t002:** An overview about the change of the scores for each class and left-out sensor. The smallest drop in accuracy is marked in green and the largest drop in red.

Left-Out Sensor	Still	Walk	Run	Bike	Car	Bus	Train	Subway	Sum
Acceleration	−14	−15	−6	−10	−23	12	−98	−25	−179
Gravity	−4	−9	−6	1	−27	−8	−86	−19	−158
Gyroscope	−24	−33	−11	−17	−37	−17	−104	−52	−295
linear Acceleration	−42	−16	−3	0	−45	−21	−101	−44	−272
Magnetometer	−205	−31	−7	−24	−66	−108	−142	−124	−707
Orientation	−48	−61	−3	−19	−53	−61	−150	−101	−496
Pressure	−123	−62	−13	−33	−67	−93	−191	−247	−829

**Table 3 sensors-21-00176-t003:** The Confusion Matrix for the private test set. The classes with the most false classifications *still* and *subway*. The classes with the best true classifications are *run* and *walk*. t = true label, p = predicted label.

t\p	Still	Walk	Run	Bike	Car	Bus	Train	Subway
still	**17,619**	75	0	28	15	75	117	71
walk	95	**17,825**	9	15	4	10	17	25
run	6	7	**17,981**	1	0	2	3	0
bike	30	39	5	**17,900**	3	9	10	4
car	11	6	1	4	**17,857**	59	44	18
bus	56	31	0	11	31	**17,711**	103	57
train	49	39	1	10	19	52	**17,685**	145
subway	52	35	0	2	21	42	274	**17,574**

**Table 4 sensors-21-00176-t004:** The difference confusion matrix of the network trained without the acceleration sensor. The largest loss in classification occurred in the class *bus* and the smallest in class *run*. t = true label, p = predicted label.

t\p	Still	Walk	Run	Bike	Car	Bus	Train	Subway
still	−14	6	2	8	2	2	−16	10
walk	−1	−15	0	0	1	5	5	5
run	4	2	−6	1	0	−3	−1	3
bike	8	7	0	−10	−4	2	1	−4
car	0	5	0	2	−23	20	−2	−2
bus	4	7	0	0	3	12	−32	6
train	17	−4	1	0	1	28	−98	55
subway	18	6	0	2	−1	18	−18	−25

**Table 5 sensors-21-00176-t005:** The difference confusion matrix of the network trained without the gravity software sensor. The largest loss in classification occurred in the class *subway* and the smallest in class *run*. t = true label, p = predicted label.

t\p	Still	Walk	Run	Bike	Car	Bus	Train	Subway
still	−4	10	0	1	7	2	−21	5
walk	−1	−9	0	−4	1	3	12	−2
run	−1	5	−6	2	0	−2	1	1
bike	−1	5	4	1	−1	−5	1	−4
car	0	2	0	2	−27	19	−2	6
bus	9	6	0	−1	8	−8	−16	2
train	13	4	0	1	12	9	−86	47
subway	11	2	0	1	5	9	−9	−19

**Table 6 sensors-21-00176-t006:** The difference confusion matrix of the network trained without the gyroscope. The largest loss in classification occurred in the class *subway* and the smallest in class *run*. t = true label, p = predicted label.

t\p	Still	Walk	Run	Bike	Car	Bus	Train	Subway
still	−24	4	1	2	12	1	−15	19
walk	−1	−33	2	0	2	12	5	13
run	−3	8	−11	1	5	−1	1	0
bike	9	4	2	−17	−1	8	−1	−4
car	7	4	1	1	−37	22	0	2
bus	5	3	0	3	21	−17	−13	−2
train	15	1	0	−1	12	11	−104	66
subway	10	6	0	0	1	21	14	−52

**Table 7 sensors-21-00176-t007:** The difference confusion matrix of the network trained without the linear acceleration software sensor. The largest loss in classification occurred in the class *subway* and the smallest in class *run*. t = true label, p = predicted label.

t\p	Still	Walk	Run	Bike	Car	Bus	Train	Subway
still	−42	7	1	9	9	22	−20	14
walk	−5	−16	4	−3	1	4	−2	17
run	−1	5	−3	3	0	−3	−1	0
bike	−2	1	0	0	0	6	−3	−2
car	−2	4	0	2	−45	35	−4	10
bus	18	3	0	3	12	−21	−14	−1
train	7	1	0	−2	17	19	−101	59
subway	18	−1	0	2	1	24	0	−44

**Table 8 sensors-21-00176-t008:** The difference confusion matrix of the network trained without the magnetometer. The largest loss in classification occurred in the class *still* and the smallest in class *run*. t = true label, p = predicted label.

t\p	Still	Walk	Run	Bike	Car	Bus	Train	Subway
still	−205	17	0	0	20	27	49	92
walk	16	−31	2	4	3	−1	2	5
run	−2	5	−7	0	0	1	2	1
bike	0	7	1	−24	−1	7	10	0
car	7	3	0	12	−66	46	−6	4
bus	12	0	0	1	28	−108	14	53
train	50	9	0	4	12	30	−142	37
subway	63	9	0	6	5	51	−10	−124

**Table 9 sensors-21-00176-t009:** The difference confusion matrix of the network trained without the orientation software sensor. The largest loss in classification occurred in the class *subway* and the smallest in class *run*. t = true label, p = predicted label.

t\p	Still	Walk	Run	Bike	Car	Bus	Train	Subway
still	−48	22	0	4	13	4	−15	20
walk	24	−61	3	−3	3	8	6	20
run	−2	5	−3	1	1	−1	−1	0
bike	18	8	0	−19	−1	−2	−1	−3
car	6	5	0	1	−53	45	−6	2
bus	18	7	0	2	11	−61	4	19
train	4	1	0	4	4	29	−150	108
subway	35	3	0	3	2	20	38	−101

**Table 10 sensors-21-00176-t010:** The difference confusion matrix of the network trained without the pressure sensor. The largest loss in classification occurred in the class *subway* and the smallest in class *run*. t = true label, p = predicted label.

t\p	Still	Walk	Run	Bike	Car	Bus	Train	Subway
still	−123	34	0	7	4	20	20	38
walk	23	−62	5	6	1	8	4	15
run	4	12	−13	0	0	−2	−1	0
bike	17	17	0	−33	5	−3	3	−6
car	3	4	0	1	−67	40	5	14
bus	17	18	0	3	19	−93	7	29
train	32	4	0	1	10	29	−191	115
subway	54	5	0	2	−3	22	167	−247

## Data Availability

Publicly available datasets were analyzed in this study. This data can be found here: http://www.shl-dataset.org/download/.

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
