# Peer review of "Analyzing the Importance of Sensors for Mode of Transportation Classificationâ€"

_sensors, 2020, doi:10.3390/s21010176_

Round 1
Reviewer 1 Report
For transportation classification, the paper evaluates the importance of each sensor data by omitting the sensor, and then comparing the results with the sensor.
The results are summarized in Table 3 and from line 308 to line 312.
Overall the paper has not provided any significant insight in terms of removing a sensor or explainability of machine learning.
It seems that removing a sensor will effect the accuracy (at least for some classes) - which is intuitive.
The paper also does not explain the black-box nature of the convolutional and the fully connected layers - as claimed on line 40.
Reviewer 2 Report
The author has subtracted the baseline confusion matrix from a confusion matrix of a network trained with an left-out sensor modality (difference confusion matrix) and visualised the low level features from the LSTM layers. This approach provides useful insights into the properties of the deep-learning algorithm and indicates the presence of redundant sensor modalities.
However, there are some questions as below:
- In the section of State of the Art, the author introduced some methods in detail, thus the reader can understand the development of research in this field. But author did not explain the advantages and disadvantages of each method and the advantages of the proposed method compared with these methods.
It is suggested that the author emphasizes several characteristic or representative methods. - In the full text, details of Figure such as zoom, coordinate axis, text size and font need to be unified. It is recommended to modify the relevant information according to the template provided from the "instruction for authors".
- I don't fully understand the details of the method. Although this paper is an extended version of a published article, the method of this time should be described in more detail in the subsection of "Algorithm" since this is the most important part.
- For facilitating the reader's reading and understanding, the order of a large number of figures and tables in the article should be adjusted according to the order of discussion. Such as Tables 1,3,4 and Figures 1,4,8. And the content of the figures should also be explained in detail.
- A large number of experimental results have been described in the section of "Discussion", but there is a lack of conclusive description.
- Although the results of the proposed method in various types of experiments have been revealed, the paper lacks comparison experiments of other methods in the same situation. It is suggested to add several typical methods as comparison and reference objects to prove the innovation and advancement of this method.
Reviewer 3 Report
In this work, the authors propose two techniques to disclose the backbox nature of deep-learning algorithms. Firstly, they assess the importance of different sensors for the mode of transportation classification through the leave-one-sensor-out strategy and then subtract the baseline confusion matrix (with all sensors) with a confusion matrix from the network trained without one sensor. Secondly, they use an autoencoder to visualize the low-level features from the LSTM layers. These techniques are applied to a model that relies on data obtained from a smartphone to classify the mode of transportation. Overall, the work proposed by the authors is interesting and the methodology used is appropriate. My comments are given as follows:
- p.1 l.24 the authors mean line-of-sight and not line of site;
- section 2. the leave-one-sensor-out strategy is not described in the state-of-art review. Why it is not described? Additionally, it would be good if the authors provide the pros and cons of each technique described and justify why they select these two techniques;
- The algorithm architecture should be better described. e.g., why do they have several "Block 2" and "Block 3" sequentially?
- the text of section 4 is quite repetitive. Some sensors have the same impact and the authors describe them separately which, in my opinion, makes the text boring to read. I would suggest the authors aggregate the sensors that have the same impact on the algorithm to improve the readability of the text.
Round 2
Reviewer 1 Report
The comments are addressed.
Author Response
Thank you for your comments.